# Aneurysm Sac Pressure during Branched Endovascular Aneurysm Repair versus Multilayer Flow Modulator Implantation in Patients with Thoracoabdominal Aortic Aneurysm

**DOI:** 10.3390/ijerph192114563

**Published:** 2022-11-06

**Authors:** Maciej Antkiewicz, Wiktor Kuliczkowski, Marcin Protasiewicz, Tomasz Zubilewicz, Piotr Terlecki, Magdalena Kobielarz, Dariusz Janczak

**Affiliations:** 1Department of Vascular, General and Transplantation Surgery, Wroclaw Medical University, 50-367 Wroclaw, Poland; 2Department of Cardiology, Wroclaw Medical University, 50-367 Wroclaw, Poland; 3Department of Vascular Surgery and Angiology, Medical University of Lublin, 20-059 Lublin, Poland; 4Department of Mechanics, Materials Science and Biomedical Engineering, Wroclaw University of Science and Technology, 50-370 Wroclaw, Poland

**Keywords:** thoracoabdominal aneurysm, aneurysm sac pressure, endovascular aneurysm repair, multilayer flow modulator

## Abstract

Open thoracoabdominal repair is the gold standard in the TAAA treatment. However, there are endovascular techniques, that sometimes may be an alternative, such as branched endovascular aneurysm repair (BEVAR) or implantation of the multilayer flow modulator (MFM). In this study, we aimed to assess differences in the aneurysm sac pressure (ASP) between patients undergoing BEVAR and MFM implantation. The study included 22 patients with TAAA (14 patients underwent BEVAR, while eight MFM implantation). The pressure sensor wire was placed inside the aneurysm. A measurement of ASP and aortic pressure (AP) was performed during the procedure. The systolic pressure index (SPI), diastolic pressure index (DPI), and pulse pressure index (PPI) were calculated as a quotient of the ASP and AP values. After the procedure, SPI and PPI were lower in the BEVAR group than in the MFM group. During a procedure, a drop in SPI and PPI was noted in patients undergoing BEVAR, while no changes were revealed in the MFM group. This indicates that BEVAR, but not MFM, is associated with a reduction in systolic and pulse pressure in the aneurysm sac in patients with TAAA.

## 1. Introduction

A thoracoabdominal aortic aneurysm (TAAA) is a type of aneurysm that is particularly difficult to treat due to its size and involvement of the renal and visceral arteries. TAAA repair requires a technique that will exclude the aneurysm from the circulation, while maintaining perfusion in the visceral and renal arteries. Open thoracoabdominal repair with the implantation of a branched graft is the gold standard in the TAAA treatment. However, the endovascular approach is a promising and modern alternative [1]. The stent-grafts used for TAAA repair have branches for extension into the visceral and renal vessel ostia, a technique called branched endovascular aneurysm repair (BEVAR) [2]. BEVAR was reported to be associated with complications including perioperative mortality in 10% of cases, spinal cord ischemia in 2.7–20% of cases, and occlusion of the target vessels (including renal arteries, celiac arteries, and superior mesenteric artery) in 5% of cases [3]. The implantation of the multilayer flow modulator (MFM) constitutes an alternative to endovascular TAAA repair. Compared with BEVAR, it is time effective and easier to perform. The MFM stents are permeable mesh constructs of cobalt alloy wires interconnected in multiple layers.

After implantation in the aorta, the MFM modulates blood flow by reducing local wall shear stress, while maintaining branch patency. The porosity rate of the stent for optimal flow modulation was determined at 65%, and stent implantation was reported to cause 90% reduction in flow velocity outside the stent [4]. However, further research is needed to assess the efficacy and safety of this method [5].

The aneurysm sac pressure (ASP) measurement after stent-graft deployment is one of the promising research directions. Previous studies show that ASP can be successfully and less invasively measured during endovascular repair [6,7] and proved the relationship between increased ASP and AAA growth or presence of endoleak (1 year after EVAR) [8,9]. Using the experimental model of endotesion, Skillern et al. concluded that high ASP after EVAR might be bounded to the need for reintervention in future [10].

In this study, we aimed to assess differences in the ASP between patients undergoing BEVAR and those subjected to MFM implantation. For this purpose, pressure measurements were performed during the procedure by means of an ultra-thin pressure guide wire used in interventional cardiology.

## 2. Materials and Methods

### 2.1. Study Design

This was a prospective experimental study based on the analysis of invasive pressure measurements obtained during endovascular TAAA repair.

### 2.2. Patients

The study included 22 patients with TAAA. The group treated with BEVAR included 14 patients (12 men and 2 women; aged 73 ± 7), while 8 patients (6 men and 2 women; aged 76 ± 4) underwent MFM implantation. The characteristic of patients is presented in Table 1.

### 2.3. Branched Endovascular Aneurysm Repair

BEVAR is performed using vascular access sites in the groin (in some cases additional axillary or brachial artery access is required). During the procedure, fluoroscopy and angiography are used to guide the accurate placement of the main stent-graft component with branches extending to the celiac trunk, superior mesenteric artery, and the right and left renal arteries. The main stent-graft component should be deployed with appropriate branches extending above the visceral and renal artery ostia. Moreover, it should be correctly placed relative to the aortic axis to ensure that the stent-graft branches are as close as possible to the ostium of the celiac trunk, superior mesenteric artery, and left and right renal arteries. After the correct deployment of the main stent-graft component, it is extended by the branched component to the iliac arteries. The ipsilateral limb of the stent-graft is extended into the common iliac artery, which ensures a complete endograft seal on this side. The contralateral limb is left without extension, with flow directed to the lower part of the aneurysm. The aim of this stage is to achieve temporary aneurysm sac perfusion through patent lumbar arteries, thus reducing the risk of spinal cord ischemia [11]. Next, using the upper extremity access, stent-grafts are deployed to connect the branches of the main stent-graft component to target renal and visceral arteries. All stent-graft components are tightened together using dedicated pressure balloons.

The second stage of the procedure is performed after 2 to 4 weeks by adding the missing component to the common iliac artery on the contralateral side. This results in complete aneurysm exclusion.

### 2.4. Multilayer Flow Modulator Implantation

In MFM (Figure 1) procedures, the venous access device is implanted via the femoral vein in the groin. First, subtraction angiography of the aorta and its branches is performed with a pigtail catheter to identify aortic lesions and landing zones. Then, a stent compressed in the deployment system is advanced into the aorta via the femoral artery access. The proximal end of the stent is deployed in the landing zone in the healthy segment of the aorta above the aneurysm. The stent is then released using the pin-and-pull delivery system. The outer sheath of the stent is pulled off, causing the stent to expand in the aortic lumen. The stent should be delivered slowly to achieve the correct position of the expanding stent relative to the anatomy of the aorta and its branches. Because the stent compressed in the deployment system is much longer than after expansion, the distal landing zone in the healthy aortic segment below the aneurysm should be carefully delineated under computed tomography angiography guidance. The main stage of MFM implantation is completed once the distal end of the stent is released from the deployment system into the distal landing zone. At the end of the procedure, follow-up angiography is performed, the endovascular device is removed, and dressing at the access site is applied.

### 2.5. Measurement Technique

In patients undergoing BEVAR, pressure measurements were done during the second stage of the procedure, before and after the iliac branch stent-graft placement using the contralateral femoral approach. In patients undergoing MFM implantation, the measurements were done before and after stent deployment.

Using local anesthesia with lidocaine, a vascular access port was implanted via the left radial artery. The port was then used to introduce a 0.014-inch pressure guide wire (Comet, Boston Scientific, Marlborough, MA, USA) with a catheter. Under fluoroscopic guidance, the guide wire was advanced to place the pressure sensor inside the aneurysm, while the distal end of the catheter was positioned in the aorta above the aneurysm. 

At each step of the procedure, a simultaneous measurement of ASP and aortic pressure (AP) above the aneurysm was performed, and the results were recorded in an electronic form. Diastolic blood pressure, systolic blood pressure, and pulse pressure were also recorded both for ASP and AP. Moreover, the systolic pressure index (SPI), diastolic pressure index (DPI), and pulse pressure index (PPI) were calculated as a quotient of the ASP and AP values. After the procedure, the guide wire and the catheter were removed under fluoroscopic guidance. The vascular access device was also removed, and the compression dressing was applied at the puncture site.

### 2.6. Statistical Analysis

For statistical analysis, the Statistica 13.3 software was used (StatSoft, Cracow, Poland). Variables were tested for normal distribution using the Shapiro-Wilk test. The t test was used for the comparison of variables between groups. The level of significance was set at a *p* value of less than 0.001.

## 3. Results

Pressure measurements were performed in all 22 patients. The presence of a guide wire in the aneurysm sac did not affect the procedure.

There were no significant differences in baseline SPI, DPI, and PPI (i.e., measured before aneurysm repair) between groups. The comparison of SPI, DPI, and PPI measurements after the procedure revealed that SPI and PPI were lower in the BEVAR group than in the MFM group (both *p* < 0.001).

Changes in SPI, DPI, and PPI before and after the procedure were also assessed within each group. The results for the BEVAR group are shown in Figure 2, and for the MFM group, in Figure 3. A drop in SPI and PPI was noted in patients undergoing BEVAR (*p* < 0.001), while no changes were revealed in patients undergoing MFM implantation.

Early outcomes (30 days after the procedure) are presented in Table 2. Aneurysm diameter in BEVAR group slightly decreased (60.5 ± 9.9 before treatment, 56.2 ± 11.7 after 30 days), while in the MFM group remains similar (69.9 ± 17.1 before treatment, 70.9 ± 17.4 after 30 days). Successful aneurysm occlusion occurred in nine of BEVAR cases. The other five cases had endoleak. The severe complication rate was much higher in the MFM group. There was one case of death due to acute intestinal ischeamia.

## 4. Discussion

Numerous authors described the use of MFM implantation for the treatment of aortic aneurysms. These were mostly preliminary studies in a limited number of patients, concluding that further research is needed to investigate the method [5,12,13,14]. A case report was published that described the emergency use of MFM for the treatment of aortic dissection, outside the indications for use [15]. The most extensive experience with the MFM technique was reported by Sultan et al. [12,16,17]. Several published papers describe findings from in-vitro [16], animal [17], and early clinical studies [12].

Lowe et al. [18] assessed mortality and side-branch vessel patency in 14 patients undergoing implantation of MFM stents for TAAAs and perirenal aneurysms involving the visceral vessels. The authors reported an all-cause mortality rate of 21%, while side-branch vessel patency was observed in 98% of cases. However, a significant increase in aneurysm diameter despite implantation and several cases of MFM dislocation requiring reintervention were also noted.

In the multicenter STRATO trial including 20 patients, Vaislic et al. [19] reported high efficacy and safety of MFM implantation, with a branch vessel patency rate of 96% at 12 months. In a follow-up report from the STRATO trial [20], the authors reported a branch vessel patency rate of 100% at 24 months and 97% at 36 months in 11 patients who continued the follow-up. No cases of stent dislocation were noted at 3 years. Moreover, there were no cases of renal or liver function deterioration. However, at 36 months, the aneurysm diameter was stable in only nine of the 20 patients (45%). Moreover, there were seven cases of death (35%) at 36 months, none of which were confirmed to be related to the aneurysm [20].

In a systematic review of 15 studies assessing the use of MFM in a total of 171 patients with complex TAAA pathology (three observational cohort studies, three multicenter cohort studies, and nine case reports), Hynes et al. [21] reported an all-cause survival rate of 53.7% at 12 months. These results are in line with our current findings. The authors concluded that further research is needed before the method becomes widely used in clinical practice [21].

Most studies on MFM treatment for TAAAs were conducted on small patient groups. The outcomes for vessel patency, mortality rates, progression of aneurysm diameter, and stent dislocation are satisfactory or at least promising.

Most authors emphasized the need for further research; however, usually, no follow-up studies were conducted [4]. In contrast, studies on the use of BEVAR and fenestrated endovascular aortic repair for TAAAs usually include many patients and report good outcomes, with an acceptable risk of complications and mortality.

In this study, we performed invasive ASP measurements to assess the perioperative efficacy of MFM implantation. To date, there have been only a few reports of ASP measurement in patients with TAAAs. One of the studies reporting ASP measurements was by Baum et al. [6] in 27 patients. Among 17 patients with endoleaks, ASP was the same as systemic pressure in 15 patients and one-half systemic pressure in two patients. Sonesson et al. [7] measured ASP in 10 patients after endovascular abdominal aortic aneurysm repair, who showed an aneurysm shrinkage of more than 6 mm at 1-year follow-up or later. They showed a marked reduction in ASP after the procedure, and almost no systolic/diastolic fluctuation, resulting in a nonpulsatile curve (PPI = 0%) [7]. Our previous research showed that ASP measurement is a safe and reliable way to determine the efficacy of treatment. Moreover, the results were correlated with an enlargement of the treated aneurysm [22].

## 5. Conclusions

Our results confirm that MFM implantation is not associated with a reduction in ASP. In each of the eight patients treated with MFM implantation, no significant fluctuations in ASP were noted. On the other hand, patients undergoing BEVAR showed a significant reduction in ASP, indicating successful aneurysm repair.

## Figures and Tables

**Figure 1 ijerph-19-14563-f001:**
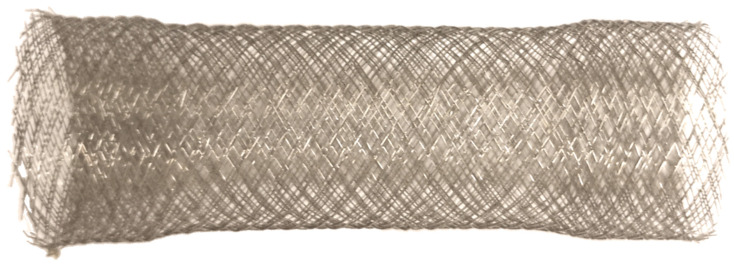
Multilayer Flow Modulator (Cardiatis SA, Isnes, Belgium).

**Figure 2 ijerph-19-14563-f002:**
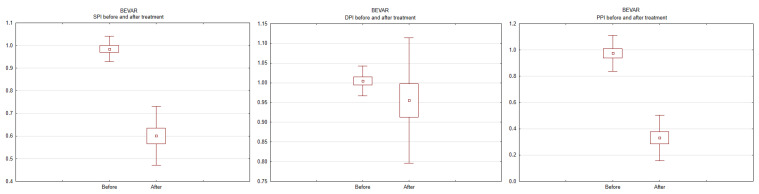
Systolic pressure index (SPI), diastolic pressure index (DPI), and pulse pressure index (PPI) at baseline and after branched endovascular aneurysm repair.

**Figure 3 ijerph-19-14563-f003:**
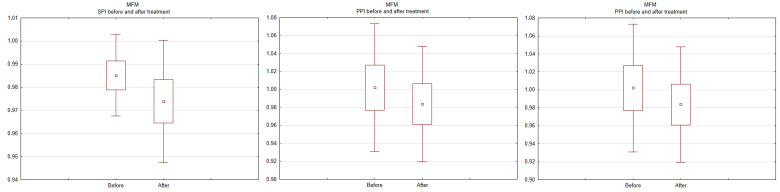
Systolic pressure index (SPI), diastolic pressure index (DPI), and pulse pressure index (PPI) at baseline and after implantation of the multilayer flow modulator. The mean SPI, DPI, and PPI values after the procedure were as follows: 0.60 ± 0.13, 0.95 ± 0.16, and 0.33 ± 0.17, respectively, for the BEVAR group, and 1.00 ± 0.03, 0.98 ± 0.12, and 1.04 ± 0.04, respectively, for the MFM group.

**Table 1 ijerph-19-14563-t001:** The clinical and demographic characteristics of patients enrolled in the study.

Variable	BEVAR	MFM
Gender		
female	2	2
male	12	6
Age (M ± SD)	73 ± 7	76 ± 4
Risk factors		
cerebral stroke	1	1
heart failure	7	3
diabetes mellitus	1	2
chronic obstructive pulmonary disease	1	2
renal insufficiency	1	2
arterial hypertension	12	6
obesity (BMI > 30 kg/cm^2^)	5	4
nicotinism	14	8
peripheral artery disease	6	4
cancer	-	2
Aneurysm diameter (mm)	60.5 ± 9.9	69.9 ± 17.1

**Table 2 ijerph-19-14563-t002:** Early outcomes (30 days after procedure).

Variable	BEVAR	MFM
Aneurysm diameter (mm)	56.2 ± 11.7	70.9 ± 17.4
Successful aneurysm occlusion	9 (64.3%)	1 (12.5%)
Visceral arteries patency	55/56 (98.2%)	30/32 (93.8%)
Severe complications		
death	-	1 (12.5%) (intestine ischemia)
myocardial infarction	-	1 (12.5%)
acute kidney injury	1 (7.1%)	2 (25.0%)
acute limb ischemia	2 (14.3%)	1 (12.5%)
graft migration	-	2 (25.0%)
Reinterventions	4 (28.6%)	3 (37.5%)

## Data Availability

The data presented in this study are available on request from the corresponding author. The data are not publicly available due to privacy restrictions.

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
