# Peer review of "Aneurysm Sac Pressure during Branched Endovascular Aneurysm Repair versus Multilayer Flow Modulator Implantation in Patients with Thoracoabdominal Aortic Aneurysm"

_ijerph, 2022, doi:10.3390/ijerph192114563_

Round 1

Reviewer 1 Report

The authors report their experience with the sac pressure measurement after TAAA endo treatment with BEVAR or MFM.

The paper suffers from many criticisms:

1. *The first 3 sentences of the abstract are uncorrect. Mention of open repair should be made. BEVAR is not still the gold standard.

2. *The conclusions of the abstract are too strong and not directly related to the results.

3. *The first period of the Introduction is not related to the article.

4. *Figure 1 cannot be put into the Introduction. In addition, please report the manufacturer of the device reported in the image.

5. *In the Introduction the authors should report data about the measurement of the sac pressure.

6. Please, do not report the Institution into the text.

7. BEVAR could be performed with femoral accesses. Axillary access is not strictly necessary.

8. *Please, remove this sentence from the Results "This indicates that BEVAR, but not MFM 3 implantation, is associated with a reduction in systolic and pulse pressure in the aneurysm 4 sac in patients with TAAA. "

Author Response

Dear Reviewer,

Thank You for such an accurate and helpful analysis. We have improved our work as suggested, especially listed critical points:

1. *The first 3 sentences of the abstract are uncorrect. Mention of open repair should be made. BEVAR is not still the gold standard.

We changed the first 3 sentences of the abstract and we made a mention of TAAA open repair.

2. *The conclusions of the abstract are too strong and not directly related to the results.

We removed strong conclusions from the abstract.

3. *The first period of the Introduction is not related to the article.

We removed first period of the Introduction.

4. *Figure 1 cannot be put into the Introduction. In addition, please report the manufacturer of the device reported in the image.

Figure 1 is now in Materials and Methods section. We added the manufacturer of the device.

5. *In the Introduction the authors should report data about the measurement of the sac pressure.

In the Introduction we reported data about the measurement of the sac pressure.

6. Please, do not report the Institution into the text.

We removed the Institution from the text.

7. BEVAR could be performed with femoral accesses. Axillary access is not strictly necessary.

We added information that BEVAR could be performed just with femoral accesses.

8. *Please, remove this sentence from the Results "This indicates that BEVAR, but not MFM 3 implantation, is associated with a reduction in systolic and pulse pressure in the aneurysm 4 sac in patients with TAAA. "

We removed  sentence from the Results.

We also added more informations about patients enrolled into the study as shor follow-up as second reviewer suggested.

We beliewe, those changes helped to improve our work and made it closer to the publication final.

Best regards,

Maciej Antkiewicz

Reviewer 2 Report

Dear editor,

Thank you for the opportunity of revising this interesting paper.

In this interesting paper, the authors compared the differences in the aneurysm sac pressure (ASP) between patients undergoing BEVAR (branched endovascular aneurysm) repair and MFM (multilayer flow modulator) implantation for a thoraco-abdominal aortic aneurysm (TAAA). The study included 22 patients with TAAA (14 patients 18 underwent BEVAR, while 8 MFM implantation). A measurement of ASP and aortic pressure (AP) was performed during procedure through a pressure sensor wire placed inside the aneurysm, while systolic pressure index (SPI), diastolic pressure index (DPI), and pulse pressure index (PPI) were calculated as a quotient of the ASP and AP values. They observed that SPI and PPI were lower in the BEVAR group than in the MFM group and a drop in SPI and PPI was noted in patients undergoing BEVAR, while no changes were revealed in MFM group. According to these results, they conclude that MFM implantation is not an effective treatment alternative in patients with TAAAs.

Major concerns are following reported:

·      MFM is supposed to modulate blood flow by reducing local wall shear stress, while maintaining branch patency. Its porosity rate for optimal flow modulation is expected to be 65%. In my opinion, you should not expect to observe a drop in aneurysm sac pressure after the implantation because with this device you are actually not excluding the aneurysm. On the other hand, it is quite obvious that you observe a drop in ASP after a BEVAR procedure, because that stent is supposed to exclude the aneurysm. Consequently, even if the pressure measurement is an interesting tool, I do not think it is enough to judge the device

·      According to literature, MFM is an alternative treatment for high-risk patients because of its simplicity and ease of deployment and its minimal invasiveness. Which was the risk profile of your MFM population? Was it comparable to the risk profile of the BEVAR group?

·      Clinical data are completely missing. Please add at least tables with pre-operative, intra-operative data of your population with differences between the BEVAR group and the MFM group

·      Population outcomes are completely missing. Please add in-hospital results and AngioCT scan results after the procedure. The MFM device actually does not promise a drop in aneurysm sac pressure, but should favor a process of aortic remodeling with thrombus deposition and stabilization of the aneurysm with visceral branches patency. Did this happen in your population?

·      To evaluate the efficacy of such a treatment, follow up data are also needed to understand the evolution of the aneurysm treated with MFM compared with standard BEVAR

·      In your conclusions, you state that “MFM implantation is not an effective treatment alternative in patients with TAAAs.” I don’t think that this paper can have such a strong conclusion on the efficacy of a device with no clinical or follow-up data available

Author Response

Dear Reviewer,

Thank You for such an accurate and helpful analysis. We have improved our work as suggested, especially listed critical points:

  • MFM is supposed to modulate blood flow by reducing local wall shear stress, while maintaining branch patency. Its porosity rate for optimal flow modulation is expected to be 65%. In my opinion, you should not expect to observe a drop in aneurysm sac pressure after the implantation because with this device you are actually not excluding the aneurysm. On the other hand, it is quite obvious that you observe a drop in ASP after a BEVAR procedure, because that stent is supposed to exclude the aneurysm. Consequently, even if the pressure measurement is an interesting tool, I do not think it is enough to judge the device
  • You are absolutely right. In our study we decided to prove a drop in ASP directly after BEVAR in contrast to MFM. It is not enough to judge the MFM, so we removed strong conclusions from our work and just focused on ASP results.
  • According to literature, MFM is an alternative treatment for high-risk patients because of its simplicity and ease of deployment and its minimal invasiveness. Which was the risk profile of your MFM population? Was it comparable to the risk profile of the BEVAR group?
  • We have added Table 1 with demographic and clinical characteristics of patients. As You mentioned, MFM is dedicated for high-risk patients and our data confirm it.
  • Clinical data are completely missing. Please add at least tables with pre-operative, intra-operative data of your population with differences between the BEVAR group and the MFM group
  • We added Table 1 with pre-operative data and Table 2 with 30-days follow up.
  • Population outcomes are completely missing. Please add in-hospital results and AngioCT scan results after the procedure. The MFM device actually does not promise a drop in aneurysm sac pressure, but should favor a process of aortic remodeling with thrombus deposition and stabilization of the aneurysm with visceral branches patency. Did this happen in your population?
  • In Table 2 we added data about sizes, aneurysm occlusion and visceral branches patency.
  • To evaluate the efficacy of such a treatment, follow up data are also needed to understand the evolution of the aneurysm treated with MFM compared with standard BEVAR
  • We collected data of short (30 days) follow up and presented them in table 2. However longer follow up as the comparment results of MFM and BEVAR treatment is topic for our further publications.
  • In your conclusions, you state that “MFM implantation is not an effective treatment alternative in patients with TAAAs.” I don’t think that this paper can have such a strong conclusion on the efficacy of a device with no clinical or follow-up data available
  • We removed this sentence from conclusion.

We beliewe, those changes helped to improve our work and made it closer to the publication final.

Best regards,

Maciej Antkiewicz

Round 2

Reviewer 1 Report

OK

Reviewer 2 Report

Dear authors,

thank you for answering to all my questions and following my suggestions on how to improve the paper. I now think it is suitable for publication. I look forward to read your further publications about the long term comparison between the two prosthesis.

Best regards